# Selected by bioinformatics and molecular docking analysis, Dhea and 2–14,15-Eg are effective against cholangiocarcinoma

Lei Qin[1], Jun Kuai[1], Fang Yang[1], Lu Yang[1], Peisheng Sun[2], Lanfang Zhang[1], Guangpeng Li [ID][3]*

1 Department of Gastroenterology, The First Affiliated Hospital of Xinxiang Medical College, Xin Xiang, China, 2 Department of Gastrointestinal Surgery, The First Affiliated Hospital of Xinxiang Medical College, Xin Xiang, China, 3 Department of Emergency, The First Affiliated Hospital of Xinxiang Medical College, Xin Xiang, China

* liguangpeng_xxyxy@163.com

## Abstract

**Data Availability Statement:** All relevant data are within the paper and its Supporting Information files.

### Object

To identify novel targets for the diagnosis, treatment and prognosis of cholangiocarcinoma, we screen ideal lead compounds and preclinical drug candidates with MYC inhibitory effect from the ZINC database, and verify the therapeutic effect of Dhea and 2–14,15-Eg on cholangiocarcinoma.

### Methods

The gene expression profiles of GSE132305, GSE89749, and GSE45001 were obtained respectively from the Gene Expression Omnibus database. The DEGs were identified by comparing the gene expression profiles of cholangiocarcinoma and normal tissues. GO, KEGG analysis and PPI network analyses were performed. LibDock, ADME and toxicity prediction, molecular docking and molecular dynamics simulations were used to identify potential inhibitors of MYC. Moreover, in vitro, MTT assay, colony-forming assay, the scratch assay and Western blotting were performed to verify the therapeutic effect of Dhea and 2–14,15-Eg.

### Results

PPI network analysis showed that ALB, MYC, APOB, IGF1 and KNG1 were hub genes, of which MYC was mainly studied in this study. A battery of computer-aided virtual techniques showed that Dhea and 2–14,15-Eg have lower rodent carcinogenicity, Ames mutagenicity, developmental toxicity potential, and high tolerance to cytochrome P4502D6, as well as could exist stably in natural circumstances. In vitro assays showed that Dhea and 2–14,15-Eg inhibited cholangiocarcinoma cellular viability, proliferation, and migration inhibiting expression of MYC.

**Funding:** The author(s) received no specific funding for this work.

**Competing interests:** The authors have declared that no competing interests exist.

## Conclusion

This study suggested that Dhea and 2–14,15-Eg were novel potential inhibitors of MYC targeting, as well as are a promising drug in dealing with cholangiocarcinoma and have a perspective application.

## Introduction

Cholangiocarcinoma (CCA) is an uncommon adenocarcinoma which arises from the epithelial cells of bile ducts that often presents with locally advanced or metastatic disease and carries an extremely poor prognosis [1]. According to the location of the pathological structure, it can be divided into three types: intrahepatic CCA (iCCA), perihilar CCA (pCCA), and distal CCA (dCCA) [2]. The current preferred treatment for CCA is surgical resection, but this method is only suitable for the early stage. For patients who in the middle and late stage do not suitable for operation, locoregional and chemoradiation treatments and targeted drug therapy are generally selected [3]. But even if comprehensive therapy is adopted, the therapeutic effect is not satisfactory. The 5-year overall survival for stages 3 and 4 CCA are 10 and 0%, respectively [4]. Besides, cumulative number of CCA deaths have been increased by 39% because of increased disease incidence. Mortality rates are higher in men and boys (1.9 per 100 000) than in women and girls (1.5 per 100 000) [5].

In recent years, bioinformatic and microarray methods become increasingly effective in exploration and analysis of multiple genes or proteins of complicated diseases [6]. By applying corresponding bioinformatics algorithms, these methods identify the core driving genes and abnormal regulatory pathways of diseases. It is helpful for researchers to reveal the therapeutic molecular targets systematically, accurately and effectively, and provide a theoretical basis for the occurrence and development of tumor. Molecular docking is an established in silico structure-based method widely used in drug discovery [7]. Virtual screening, a computational technique with a diverse set of available tools [8], can select active compounds with drug properties from millions of molecules by molecular docking. Therefore, Virtual screening and molecular docking are extensively practical method in rational drug design and medicinal chemistry [9, 10]. For example, several new drugs for advanced diseases have been developed, including FGFR inhibitors and IDH inhibitors, aiming at the potential driver genetic aberrations in CCA [11].

In this study, we use the combination of bioinformatics and virtual screening methods to sift drugs that can bind to specific targets, so as to promote the research and development of cholangiocarcinoma drugs. Besides, this method has proven to be highly effective and has contributed to the treatment of other diseases, such as osteosarcoma and glioblastoma [12, 13]. In current study, 3 mRNA microarray datasets (GSE132305, GSE89749, and GSE45001) involving CCA were downloaded from Gene Expression Omnibus database, and those datasets were analyzed to identify differentially expressed genes (DEGs) by comparing gene expression profiles of the CCA and normal tissues. Then, the mutual DEGs were screened with a Venn analysis. Gene Ontology (GO) and Kyoto Encyclopedia of Genes and Genomes (KEGG) enrichment analysis were performed to study alterations in biological functions and signaling pathways of CCA. PPI network construction was performed, followed by identification of hub genes. Next, a range of structural biology methods, involving virtual screening, molecular docking and so on, were used to screen and identify lead compounds with potential inhibitory effects on MYC. In addition, our study also predicted the absorption, distribution, metabolism,

excretion (ADME) and toxicity of these compounds. This study provides a novel medication candidate for CCA treatment.

## Materials and methods

### Microarray data

The gene expression profiles of GSE132305, GSE89749, and GSE45001 were obtained from Gene Expression Omnibus database (http://www.ncbi.nlm.nih.gov/geo) [14]. The corresponding profiles were provided on platform GP13667 (GSE132305), GPL10558 (GSE89749) and GPL14550 (GSE45001), containing a total of 310 samples of CCA and 50 normal samples. GSE132305 included 182 CCA samples and 38 normal samples, GSE89749 provided 118 CCA samples and 2 normal samples, and GSE45001 contained 10 CCA samples and 10 normal samples.

### Identification of DEGs

The analyses of the raw data were conducted using R Language for three groups of DEGs to fit three respective gene expression profiles. Hierarchical cluster analysis was used to classify the data and identify the CCA group and normal samples. Principal component analysis was used to determine the probe quality control in Genespring, and probes with intensity values below the 20th percentile were filtered out using the "filter probesets by expression" option. Then, the DEGs were identified using a classic $t$ test with a P value cutoff of <0.05 and a change >1 fold, which were applied for a statistically significant definition. We also computed Venn diagrams for upregulated, downregulated, and total DEGs.

### GO and pathway enrichment analysis of DEGs

The DAVID database(Database for Annotation, Visualization and Integrated Discovery, http://david.abcc.ncifcrf.gov/) is an online program that provides researchers with a comprehensive set of functional annotation tools to understand the biological significance behind a large number of genes [15]. Gene Ontology (GO) is a useful method for biological process (BP), cellular component (CC), and molecular function (MF) of genes. Kyoto Encyclopedia of Genes and Genomes(KEGG) is involved in gene function annotation, as well as the basis of analysis and genomic information link. Metascape (https://metascape.org/gp/index.html) is a dependable, intuitive tool not only for pathway enrichment and biological process annotation, but also for the analysis of gene-related protein networks and related drugs [16].

### Protein-Protein Interaction (PPI) network construction and module selection

To uncover the hub genes, we used the STRING (Search Tool for Retrieval of Interacting Genes, http://string.embl.de/) database [17] to construct a PPI network. These interaction networks were visualized using Cytoscape [18]. Then, revealing modules of the PPI network with MCODE (Molecular Complex Detection) [19]. In addition, the function and pathway of DEGs in the module were enriched and analyzed. This process is to identify the HUB gene and its degree.

### Discovery studio software and ligand libraries

Discovery studio, a set of software for molecular modeling and simulation environment, aims to screen, design and modify potential drugs through structural chemistry and structural biology calculation, so as to identify and refine a wide range of lead compounds and candidate

drugs [8]. Virtual screening is performed using the LibDock and ADME (absorption, distribution, metabolism, excretion) modules of Discovery Studio 4.5 software (DS4.5, Accelrys, Inc.). CDOCKER is applied for docking research. ZINC15 database was used to screen MYC inhibitors [20].

### Use LibDock for structure-based virtual filtering

The ligand-binding pocket region of MYC was selected to identify new compounds that might inhibit MYC as the binding site. The LibDock module of Discovery Studio 4.5 was used in Virtual filtering [21]. LibDock, a rigid docking program, uses grids placed at binding sites and polar and non-polar probes to compute protein hotspots. In order to form well interactions, the hotspots are further used to align ligands, as well as the Smart Minimiser algorithm and CHARMm force field (Cambridge, Massachusetts, USA) are used to minimize ligands. All the minimized ligand positions were ranked according to the ligand scores. The crystal structure of MYC was downloaded from the protein database (PDB) and imported into the working environment of LibDock. Proteins are made by removing crystal water and other heteroatoms, adding hydrogen, and then protonation, ionization and energy minimization. In order to achieve energy minimization, CHARMM force field and Smart Minimiser algorithm were usually used [22]. All the prepared ligands were docked on the defined active sites for virtual screening by using LibDock. All docking locations are sorted and grouped by composite name According to the LibDock score.

### ADME (Absorption, Distribution, Metabolism, and Excretion) and toxicity prediction

Interactions between ligands and protein were analyzed and visualized in Discovery Studio 4.5 (DS4.5). The ADME module of DS4.5 is used to calculate the absorption, distribution, metabolism, and excretion of selected compounds, also used the DS4.5 TOPKAT (toxicity prediction by Computer assistive technology) module to calculate all potential compounds toxicity and other properties, including its aqueous solubility, blood-brain barrier (BBB) permeability, cytochrome P4502D6 (CYP2D6), hepatotoxicity, human intestinal absorption, plasma protein (PPB) level, rodent carcinogenicity, Ames respectively and developmental toxicity potential. These pharmacological properties were fully considered in the selection of MYC drug candidates.

### Molecule docking

The obtained compounds were advanced to molecular docking employing the CDOCKER module available with Discovery Studio (DS) [23]. CDOCKER is a molecular docking method based on CHARMM, which uses high temperature dynamics to search the flexible conformation space of ligand molecules, and uses the CHARMm energy and interaction energy to indicate the ligand-binding affinity, so as to make the docking results more accurate. During rigid and semi-flexible docking processes, we usually remove the crystal water molecules and add hydrogen atoms to reduce the negative effect of immobilized water molecules on receptor-ligand complex formation [24]. The ligands were permitted to attach to residues within binding site spheres during the docking process. The initial compound was extracted from the binding site and then realigned into the crystalline structure of MYC to demonstrate the reliability of the combination pattern. The CDOCKER Energy was used to analyze the different postures of each tested molecule. The ligand pose corresponding to the highest score was taken as the best-docked pose [3].

## Molecular dynamics simulation

The best binding conformations of each compounds-MYC complex were selected and prepared for molecular dynamics simulation. The ligand-receptor complex was placed in an orthogonal box and solved by an explicit periodic boundary solvated water model. In order to simulate the physiological environment, silicon chloride with ionic strength of 0.145 was added into the system. Next, the system was subjected to the CHARMm forcefield and relaxed by energy minimization (500 steps of steepest descent and 500 steps of conjugated gradient), with the final root means square gradient of 0.227. In 2ps, the system was slowly driven from the initial temperature of 50K to the target temperature of 300K, and the equilibrium simulation was carried out for 5ps. Molecular dynamics simulations (production) were run for 25ps with a time step of 1fs. The simulation was performed with the normal pressure and temperature system at a identical temperature of 300K. Long-range electrostatics were calculated by the particle mesh Ewald algorithm, and all bonds involving hydrogen were fixed by the linear constraint solver algorithm. Taking the initial complex setting as a reference, the trajectory was determined based on structural properties, root mean-square deviation (RMSD), and potential energy by using trajectory protocol in Discovery Studio 4.5.

## MTT assay

The cholangiocarcinoma cells (HuCCT1) were inoculated into 96-well plates with a density of 500 cells/well, and different doses of Dhea and 2–14,15-Eg were injected into the plates. The MTT reagent(Sigma, St. Louis, Missouri, USA) was dissolved in phosphate-buffered saline (PBS, 5 mg/mL) to measure the viability of cells. On the day of measurement, fresh DMEM supplemented with 10% fetal bovine serum and diluted MTT (1:10, 10% MTT) were used to replace the medium, and incubated at 37°C for 3.5 h. Then the incubation medium was taken out and the formalin crystal was dissolved in 200μsolution of DMSO. Last, we used an ELx800 absorbance microplate reader (BioTek Instruments, VT, USA) to quantify the MTT reduction by measuring light absorbance at 570 nm.

## Colony-forming assay

The cholangiocarcinoma cells (HuCCT1) were seeded in a dish with a density of 50 cells/cm2. After 24 hours of culture, the cells were treated with different doses of Dhea and 2–14,15-Eg. After 10 days of culture in vitro, the colonies were counted and described according to Franken et al. The colonies were washed with PBS, fixed with 4% paraformaldehyde, dyed with 5% crystal violet for 0.5 h, and washed with water twice.

## In vitro scratch assay

The cholangiocarcinoma cells (HuCCT1) were cultured on 24 well permanox plates. Use a 1 ml pipette tip in each well to create a consistent cell-free area where the loose cells were gently washed out with DMEM. After that, the cells were exposed to different doses of Dhea and 2–14,15-Eg. The images of the scratched area were captured by phase contrast microscope at 0 and 24 hours after the scratch. The residual damage area and scratch width at 6 different points of each image were measured.

## Western blotting

The cholangiocarcinoma cells (HuCCT1) were inoculated into 96-well plates with a density of $2 \times 105$ cells/well, and different doses of Dhea and 2–14,15-Eg were injected into the plates. After 48 h, total proteins were harvested and electrophoretically separated. The proteins were

transferred to membranes, which were treated with primary antibodies against MYC and GAPDH and then incubated with secondary antibodies. The membranes were visualized with an enhanced chemiluminescence detection system (Pierce; Thermo Fisher Scientific, Inc.).

## Results

### Identification of DEGs

An aggregate of 301 DEGs were identified from GSE132305, of which 165 were upregulated and 136 were downregulated. A total of 2436 DEGs were identified in GSE89749, among which 2204 were upregulated and 232 were downregulated. A total of 3445 DEGs were picked up from GSE45001, which 1527 genes were upregulated and 1918 genes were downregulated (**Figs 1A** and S1A). By performing Venn diagram, a total of 492 mutual DEGs were identified among these 3 datasets (**Fig 1B**).

### Functional and pathway enrichment analysis

To further explore the function of identified DEGs, the mutual upregulated and downregulated DEGs were entered into DAVID for GO and KEGG pathways analysis (**Fig 2A and 2B and S1 Table**). The GO analysis results showed that the mutually upregulated DEGs were mainly associated with several biological processes (BPs), such as metabolic process, positive regulation of smooth muscle cell proliferation and cell surface receptor signaling pathway; cellular components (CCs), like extracellular exosome, extracellular space and extracellular region; and molecular Functions (MFs) covering heparin binding, transcriptional activator activity and glutathione transferase activity. For the mutual downregulated DEGs, the GO analysis results that were primarily enriched in BPs including extracellular matrix organization and cell adhesion; CCs involving cytoplasm and nucleoplasm; and MFs containing protein binding and structural molecule activity. KEGG analyses demonstrate the most significant enriched pathways of the mutual DEGs, for example, chemical carcinogenesis, fat digestion and absorption and drug metabolism—cytochrome P450. In addition, the metascape results showed the enrichment in extracellular structure organization, fat digestion and absorption, lipid transport and et al (**Figs 2C, 2D** and S1B). The results of GSEA analysis and KEGG analysis indicated that the expression profiles of CCA were mainly enriched in "P53 signaling pathway" and "adipocytokine signaling pathway" (**Fig 2E**).

### Hub genes and modules screening from the PPI network

We also conducted PPI network analyses of the previous 492 mutual DEGs. Genes $\geq 20$ degrees were screened as hub genes based on the STRING database. A total of 20 genes were identified as hub genes: ALB, MYC, APOB, IGF1, KNG1, FOS, CCL2, SPP1, COL1A1, THBS1, EGR1, HP, APOA4, ORM1, MMP1, TF, APOC3, FGB, APOA2, ITGA2, KLF4 (**Table 1**). Among these genes, 421 nodes and 1530 edges were obtained, the node degree of ALB was the highest, followed by MYC.

Besides, after MCODE analysis, a total of 14 modules were generated, the top three modules were selected as well (S2 Table, **Fig 3**). The functional annotation and enrichment of modules genes were shown in S2 Table. Enriched function analysis indicated that genes in module 1 were associated with extracellular space, extracellular exosome, and extracellular region. In module 2, the genes were mainly related in drug metabolism—cytochrome P450, chemical carcinogenesis, and metabolism of xenobiotics by cytochrome P450. At last, for module 3, the genes were enriched in fatty acid degradation, fatty acid metabolism, and metabolic pathways.

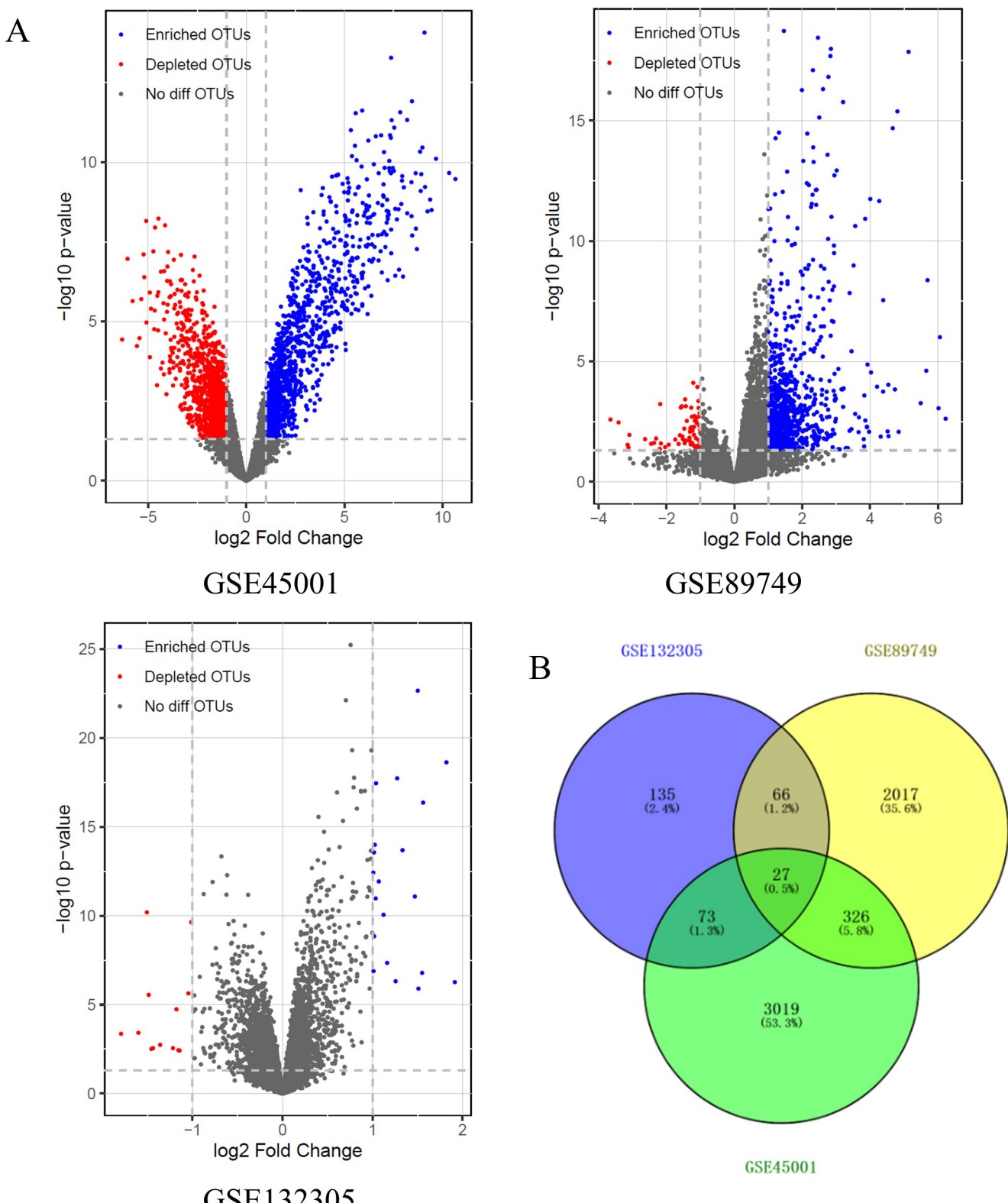

**Fig 1.** Identification of DEGs (A) volcanic plot of the three datasets (B) Venn plot of DEGs among the three datasets.

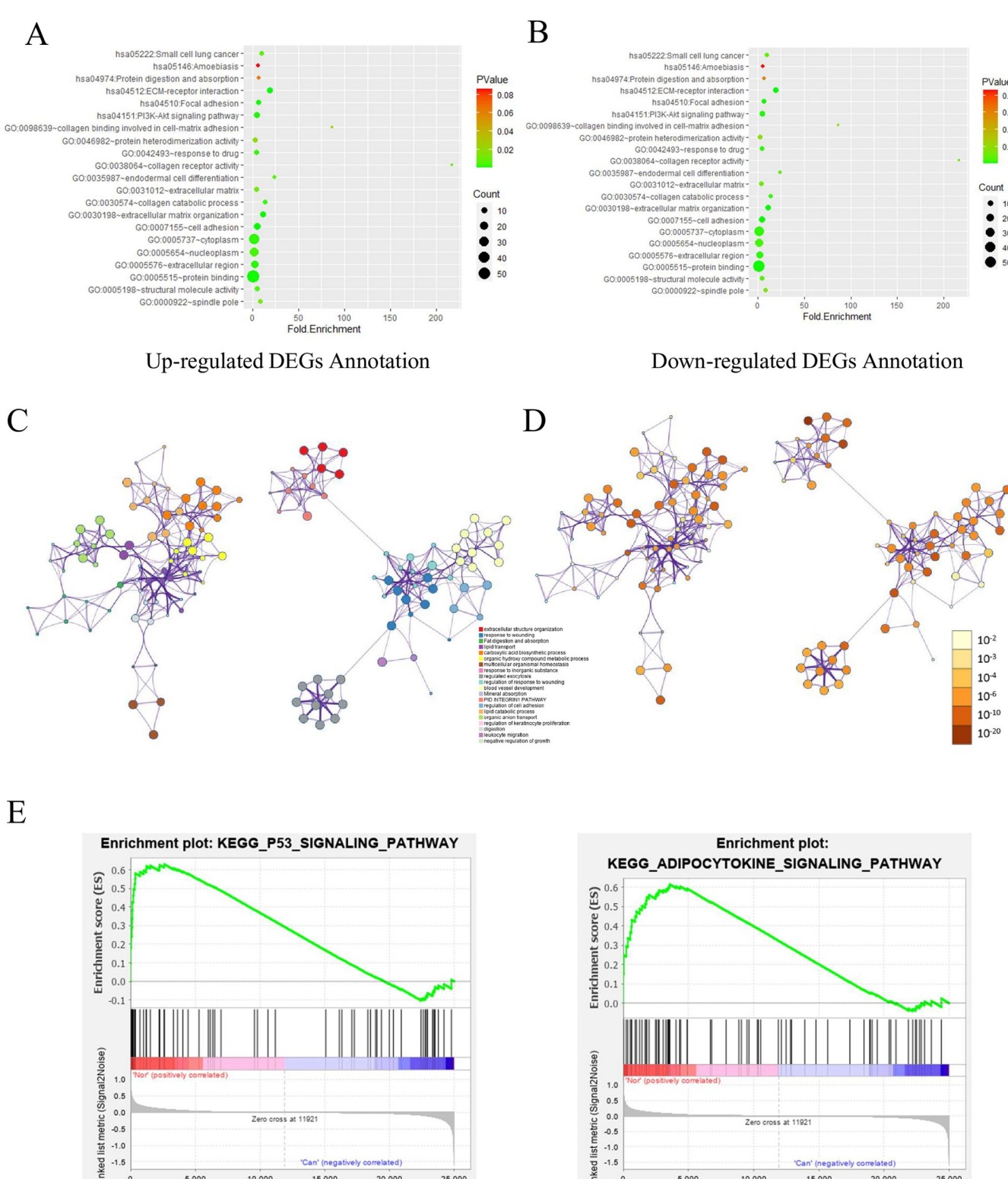

**Fig 2.** Functional and Pathway Enrichment Analysis (A) Functional annotation and enrichment of up-regulated DEGs. (B) Functional annotation and enrichment of down-regulated genes. (C) colored by cluster ID, where nodes that share the same cluster ID are typically close to each other in DEGs. (D) colored by p-value, where terms containing more genes tend to have a more significant P-value in DEGs. (E) GSEA analysis results.

**Table 1. Detailed information of the hub genes.**

| Gene symbol | Degree | Betweenness Centrality | Gene symbol | Degree | Betweenness Centrality |
|---|---|---|---|---|---|
| ALB | 96 | 0.32568879 | EGR1 | 26 | 0.03259314 |
| MYC | 50 | 0.13048765 | HP | 25 | 0.01071026 |
| APOB | 42 | 0.04473861 | APOA4 | 25 | 0.0120723 |
| IGF1 | 39 | 0.04056609 | ORM1 | 25 | 0.00941632 |
| KNG1 | 38 | 0.04386368 | MMP1 | 23 | 0.02351919 |
| FOS | 36 | 0.04414542 | TF | 22 | 0.00762613 |
| CCL2 | 34 | 0.04374119 | APOC3 | 21 | 0.00742394 |
| SPP1 | 32 | 0.0280843 | FGB | 21 | 0.00825629 |
| COL1A1 | 30 | 0.03390233 | APOA2 | 20 | 0.00239813 |
| THBS1 | 29 | 0.03833588 | ITGA2 | 20 | 0.0260957 |

## Virtual screening of small molecules inhibitors

MYC is an oncogene and is considered to be a primary hub gene in the development of CCA. Therefore, a virtual screening has been conducted to identify the most effective small molecule inhibitors against MYC. A sum of 17931 ligands were screened from the ZINC15 database. After calculated by LibDock, top 20 ranked molecules with LibDock scores were listed in **S3 Table.**

## ADME and toxicity prediction

The pharmacologic properties of the whole selected ligands included solubility level, blood-brain barrier (BBB), CYP2D6 binding, hepatotoxicity, human intestinal absorption and plasma protein (PPB) level were predicted by ADME module of Discovery Studio 4.5 (**Table 2**). According to aqueous solubility prediction, all but three compounds are soluble in water. There are undefined levels about the blood-brain barrier for most of the compounds, except ZINC000008689961, ZINC000027646625, ZINC000013838499 and ZINC000000900047. Seven tenths of the compounds were predicted to be non-inhibitors CYP2D6, which had a great influence on drug metabolism. In terms of hepatoxicity, half of compounds were found to be nontoxic and the rest were toxic. As to human intestinal absorption, 9 compounds were predicted to have the good absorption. Plasma protein binding properties indicated 6 compounds had weak absorption.

Safety should be highly considered in the research process. To ensure the safety of these 20 compounds, the toxicity indexes of these compounds, including developmental toxicity potential properties, rodent carcinogenicity (based on the U.S. National Toxicology Program dataset), as well as Ames mutagenicity, were predicted by using the computational method in the TOPKAT module (**Table 3**). The results indicate that 5 compounds were found to be non-mutagenic, and 9 compounds were found with no developmental toxicity potential. In summary of all the above results, ZINC000008689961 and ZINC000027646625 were determined to be the perfect lead compounds. Compared with other compounds, they were non-CYP2D6 inhibitors, and had no hepatotoxicity, lower Ames mutagenicity, developmental toxicity potential as well as rodent carcinogenicity. On the whole, ZINC000008689961 (Dhea) and ZINC000027646625 (2–14,15-Eg) were considered as safe drugs and chosen for the following study (**Fig 4A, 4B and 4C**).

## Analysis of ligand binding

We docked ZINC000008689961 and ZINC000027646625 into the molecule structure of MYC by CDOCKER module, so as to study ligand blinding mechanisms of these compounds with

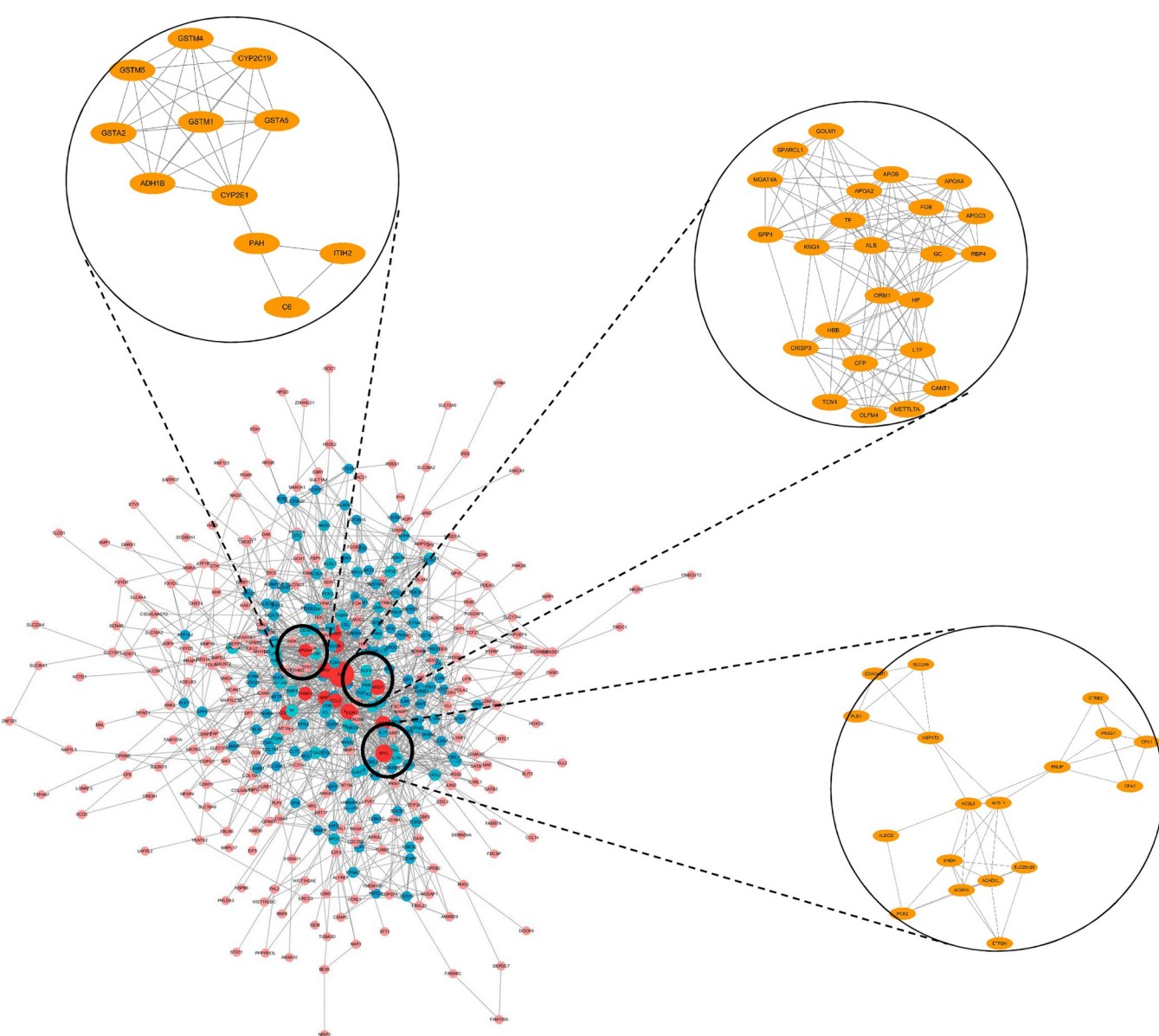

**Fig 3. Top three modules from the protein-protein interaction network.** The degrees of nodes were labeled in different colors, the red color means bigger degrees.

MYC. And then, the CDOCKER potential energy was calculated and displayed as shown in **Table 4**. The result showed that MYC may have high binding affinity with ZINC00008220033 and ZINC00001529323. In addition, we analyzed the Hydrogen bonds interaction and performed it through a structural computation (**Figs 4D–4G,** S2A and S2B). Outcomes illustrated that 4 pairs of hydrogen bonds of ZINC000008689961 with MYC were formed and ZINC000027646625 formed 7 hydrogen bonds with MYC (**S4 Table**).

## Molecular dynamics simulation

In order to evaluate the stabilities of ligand-MYC complexes in natural environment, a molecular dynamics simulation module was established. Through molecular docking experiment, the

**Table 2. Adsorption, distribution, metabolism, and excretion properties of compounds.**

| Number | Compounds | Compounds name | Solubility Level | BBB Level | CYP2D6 | Hepatotoxicity | Absorption Level | PPB Level |
|---|---|---|---|---|---|---|---|---|
| 1 | ZINC000028968107 | Cochinchinenin C | 1 | 4 | 1 | 1 | 3 | 1 |
| 2 | ZINC000005158610 | 2',4',5,7-Tetrahydroxy-5'-Geranylflavanone | 2 | 4 | 1 | 0 | 2 | 1 |
| 3 | ZINC000014946303 | Cellulose Triacetate | 0 | 4 | 0 | 1 | 3 | 0 |
| 4 | ZINC000013838499 | Guineensine | 2 | 0 | 0 | 0 | 1 | 1 |
| 5 | ZINC000040164463 | Cer | 2 | 4 | 0 | 0 | 3 | 0 |
| 6 | ZINC000002509755 | O-Desmethylcarvedilol | 2 | 2 | 1 | 1 | 0 | 1 |
| 7 | ZINC000002126785 | Umbelliprenin | 1 | 4 | 0 | 0 | 3 | 1 |
| 8 | ZINC000000900047 | Silica Aerogel | 2 | 1 | 0 | 1 | 0 | 1 |
| 9 | ZINC000008689961 | Dhea | 3 | 0 | 0 | 0 | 0 | 0 |
| 10 | ZINC000027646625 | 2–14,15-Eg | 3 | 1 | 0 | 0 | 0 | 0 |
| 11 | ZINC000004098466 | Vismione D | 2 | 4 | 0 | 0 | 1 | 1 |
| 12 | ZINC000002526388 | 4'-Hydroxycarvedilol | 2 | 4 | 1 | 1 | 0 | 1 |
| 13 | ZINC000004655034 | Zeta1-Tocopherol | 0 | 4 | 0 | 0 | 3 | 1 |
| 14 | ZINC000002526389 | 5'-Hydroxycarvedilol | 2 | 4 | 1 | 1 | 0 | 1 |
| 15 | ZINC000004654839 | Demethylphylloquinone | 0 | 4 | 0 | 0 | 3 | 1 |
| 16 | ZINC000001667453 | epinortrachelogenin | 3 | 4 | 0 | 0 | 0 | 1 |
| 17 | ZINC000002528509 | 4'-Hydroxycarvedilol | 2 | 4 | 1 | 1 | 0 | 1 |
| 18 | ZINC000002097863 | Nitrobenzylmercaptopurine Ribonucleoside | 3 | 4 | 0 | 1 | 3 | 0 |
| 19 | ZINC000015206004 | Murrayanol | 1 | 4 | 0 | 1 | 3 | 1 |
| 20 | ZINC000001318428 | N6-Benzyladenosine | 4 | 4 | 0 | 1 | 0 | 0 |

BBB, blood-brain barrier; CYP2D6, cytochrome P-450 2D6; PPB, plasma protein binding

Aqueous-solubility level: 0, extremely low; 1, very low, but possible; 2, low; 3, good.

BBB level: 0, very high penetrant; 1, high; 2, medium; 3, low; 4, undefined.

CYP2D6 level: 0, noninhibitor; 1, inhibitor.

Hepatotoxicity: 0, nontoxic; 1, toxic.

Human-intestinal absorption level: 0, good; 1, moderate; 2, poor; 3, very poor.

PPB: 0, absorbent weak; 1, absorbent strong.

original conformations were obtained by CDOCKER module. RMSD curves and potential energy diagram of the complexes were shown in **Fig 4H and 4I**. After 22 ps, the trajectories of each complex reached equilibrium. As time goes on, RMSD and potential energy of these complexes gradually stabilize. Molecular dynamics simulations confirmed that the hydrogen bond and P-dependent interactions between the compound and MYC were contributed to the stability of the complex. In conclusion, ZINC000008689961 and ZINC000027646625 can work together with MYC, and the complexes exist stably in the natural environmental circumstances which had an influence on MYC.

## Dhea and 2–14,15-Eg reduces proliferation of CCA cells

In order to evaluate the sensitivity of CCA cells to Dhea and 2–14,15-Eg, the number of survival cells treated with Dhea and 2–14,15-Eg were measured by MTT assay. As shown in **Fig 5A and 5B**, the survival rate of HuCCT1 was decreased significantly with the increase of drug concentration. To further determine the anticancer effect of Dhea and 2–14,15-Eg in CCA cells, we carried out colony-forming assay. The results suggested that the incidence of clone formation in the culture dish using Dhea and 2–14,15-Eg was fewer and smaller than that in

**Table 3. Toxicities of compounds.**

| Number | Compounds | Compounds name | Mouse NTP | | Rat NTP | | Ames | DTP |
|---|---|---|---|---|---|---|---|---|
| | | | Female | Male | Female | Male | | |
| 1 | ZINC000028968107 | Cochinchinenin C | 1 | 0.021 | 0.06 | 0.997 | 1 | 1 |
| 2 | ZINC000005158610 | 2',4',5,7-Tetrahydroxy-5'-Geranylflavanone | 0 | 1 | 1 | 0 | 1 | 1 |
| 3 | ZINC000014946303 | Cellulose Triacetate | 0.991 | 1 | 0 | 1 | 0 | 0.007 |
| 4 | ZINC000013838499 | Guineensine | 0.193 | 1 | 0 | 0 | 1 | 0.968 |
| 5 | ZINC000040164463 | Cer | 0.994 | 0.43 | 0 | 0.999 | 0.001 | 0 |
| 6 | ZINC000002509755 | O-Desmethylcarvedilol | 0.603 | 0.001 | 0 | 0.535 | 0.996 | 0.019 |
| 7 | ZINC000002126785 | Umbelliprenin | 0 | 1 | 1 | 0.002 | 0.865 | 0 |
| 8 | ZINC000000900047 | Silica Aerogel | 0.177 | 0 | 0 | 0.03 | 0.319 | 0 |
| 9 | ZINC000008689961 | Dhea | 0 | 0 | 0 | 0 | 0 | 0 |
| 10 | ZINC000027646625 | 2–14,15-Eg | 0 | 0 | 0 | 0 | 1 | 0 |
| 11 | ZINC000004098466 | Vismione D | 0.09 | 1 | 1 | 1 | 1 | 1 |
| 12 | ZINC000002526388 | 4'-Hydroxycarvedilol | 0.999 | 0.041 | 0 | 0.999 | 0.999 | 0.745 |
| 13 | ZINC000004655034 | Zeta1-Tocopherol | 0 | 1 | 1 | 0 | 1 | 1 |
| 14 | ZINC000002526389 | 5'-Hydroxycarvedilol | 0.999 | 0.036 | 0 | 0.999 | 0.999 | 0.769 |
| 15 | ZINC000004654839 | Demethylphylloquinone | 1 | 1 | 0 | 0.994 | 0.019 | 1 |
| 16 | ZINC000001667453 | epinortrachelogenin | 0.868 | 0 | 0.942 | 0.992 | 0.706 | 1 |
| 17 | ZINC000002528509 | 4'-Hydroxycarvedilol | 0.999 | 0.041 | 0 | 0.999 | 0.999 | 0.745 |
| 18 | ZINC000002097863 | Nitrobenzylmercaptopurine Ribonucleoside | 1 | 1 | 0 | 0 | 1 | 0 |
| 19 | ZINC000015206004 | Murrayanol | 0.999 | 1 | 1 | 0.005 | 0.004 | 1 |
| 20 | ZINC000001318428 | N6-Benzyladenosine | 1 | 1 | 0 | 0 | 1 | 0.003 |

NTP,U.S.NationalToxicologyProgram;DTP,developmentaltoxicitypotential.

NTP<0.3(noncarcinogen);>0.8(carcinogen).

Ames<0.3(nonmutagen);>0.8(mutagen).

DTP<0.3(nontoxic);>0.8(toxic).

the control group (**Fig 5C and 5D**), and the effect of Dhea was significantly better than that of 2–14,15-Eg.

## Dhea and 2–14,15-Eg reduces migration of CCA cells

The effect of Dhea and 2–14,15-Eg on invasion and migration of CCA cells was verified by scratch assay in vitro. The width of scratch area was recorded after scratch and 24 hours later. As shown in **Fig 5E and 5F**, the scratch width of the control group decreased significantly after 24 hours, while that of Dhea group and group 2–14,15-Eg decreased slightly. In addition, with the passage of time, the wounds in the control group were also significantly smaller than in the drug group.

## Dhea and 2–14,15-Eg reduces MYC expression in CCA cells

To verify that the effects of Dhea and 2–14,15-Eg were due to its inhibition of MYC in CCA cells, we assessed MYC levels using Western blotting. Results demonstrated that MYC expression decreased with increasing drug concentrations (**Fig 5G and 5H**). These findings suggested that Dhea and 2–14,15-Eg kill CCA cells by inhibiting MYC.

## Discussion

CCA, the second primary liver cancer, is usually diagnosed at an unresectable advanced stage. After disease progression, there are few treatment options for gemcitabine and cisplatin first-

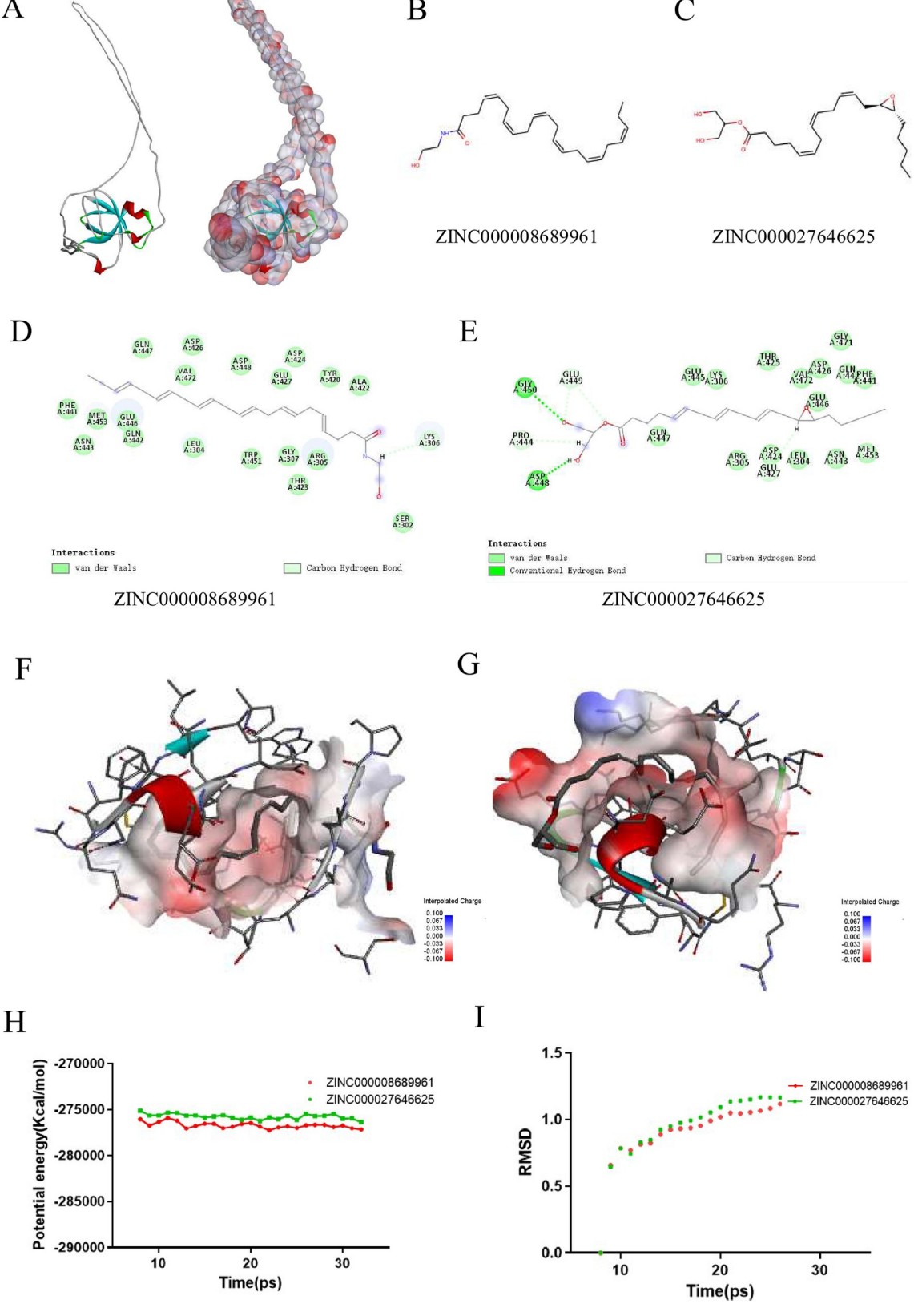

**Fig 4.** Virtual screening of small molecules inhibitors (A) The crystal structure of MYC. (B) Schematic of intermolecular interaction of the predicted binding modes of ZINC000008689961 and (C) ZINC000027646625. (D) Schematic of intermolecular interaction of the predicted binding modes of ZINC000008689961 with MYC, and (E) ZINC000027646625 with MYC. (F)(G) The docking detail between compounds with MYC (H) Potential energy profiles and (I) RMSD of compounds ZINC000008689961 and ZINC000027646625 performed by molecular dynamic simulation.

line chemotherapy, resulting in a poor prognosis [25]. Over the past four decades, the incidence of CCA has been increasing. The increase in CCA mortality was due to the anatomical location and growth pattern of CCA and the lack of clear diagnostic criteria, the diagnosis is difficult [26]. In recent years, studies on CCA have been reported, but its pathogenesis remains unclear and needs further study. Therefore, this study provides a promising target for the treatment of CCA, which is of great importance for its diagnosis, treatment and prognosis.

In the present study, we analyzed the gene expression profiles of 310 CCA samples and 50 normal samples from messenger RNA microarray datasets GSE132305, GSE89749, and GSE45001 in the GEO database. From these 3 datasets, 301 DEGs, 2,436 DEGs, and 3,445 DEGs were respectively identified, as well as a total of 492 mutual DEGs were screened.

GO analysis of abnormally expressed genes showed that upregulated genes were mainly associated with metabolic process, positive regulation of smooth muscle cell proliferation, cell surface receptor signaling pathway and extracellular exosome. Previous studies have shown that exosome mediated factors can promote tumor initiation, metastasis and drug resistance through intercellular communication [27]. Therefore, exosomes can be used as a promising diagnostic source of cancer [28]. The downregulated genes were mostly associated with extracellular matrix organization, cell adhesion, protein binding and structural molecule activity, which may be related to the fast multiplication of cancer cells. According to the results of KEGG analysis, the mutual DEGs were mainly involved in P53 signaling pathway, adipocytokine signaling pathway, chemical carcinogenesis, fat digestion and absorption and drug metabolism-cytochrome P450. The P53 signaling pathway is involved in apoptosis, growth inhibition, inhibition of cell cycle progression and acceleration of DNA repair [29]. Therefore, p53 is mutated in the vast majority of tumor cells and in more than 50% of all malignant tumors. Also, obesity could account for up to 20% of cancer-related deaths. The association is due to the metabolic and inflammatory changes of adipose tissue, which destroy the physiological homeostasis of local tissues and systems [30]. This suggests that obesity may increase the risk of CCA.

In order to obtain the hub genes among the identified DEGs, 492 mutual DEGs were analyzed with the PPI network base on the STRING database. A total of 21 genes were screened with degrees ≥ 20, particularly in ALB, MYC, APOB, IGF1 and KNG1. MYC, a transcription factor, plays an important role in regulating cellular processes, such as, survival, proliferation, metabolism, and signal transduction to control of DNA replication. It is also a oncogene. If MYC is overexpressed due to genetic factors or other mechanisms, it can lead to canceration in normal tissues, which is also the pathogenesis of many tumors, including various solid tumors and lymphoid malignancies [31]. In this study, we mainly explored its relationship with CCA. A previous research suggests that tumors with higher versus lower MYC protein expression should more aggressive, and hence that MYC protein expression should be

**Table 4. CDOCKER potential energy of compounds with MYC.**

| Compound | -CDOCKER Potential Energy (kcal/mol) |
|---|---|
| ZINC000008689961 | 28.1454 |
| ZINC000027646625 | 34.2787 |

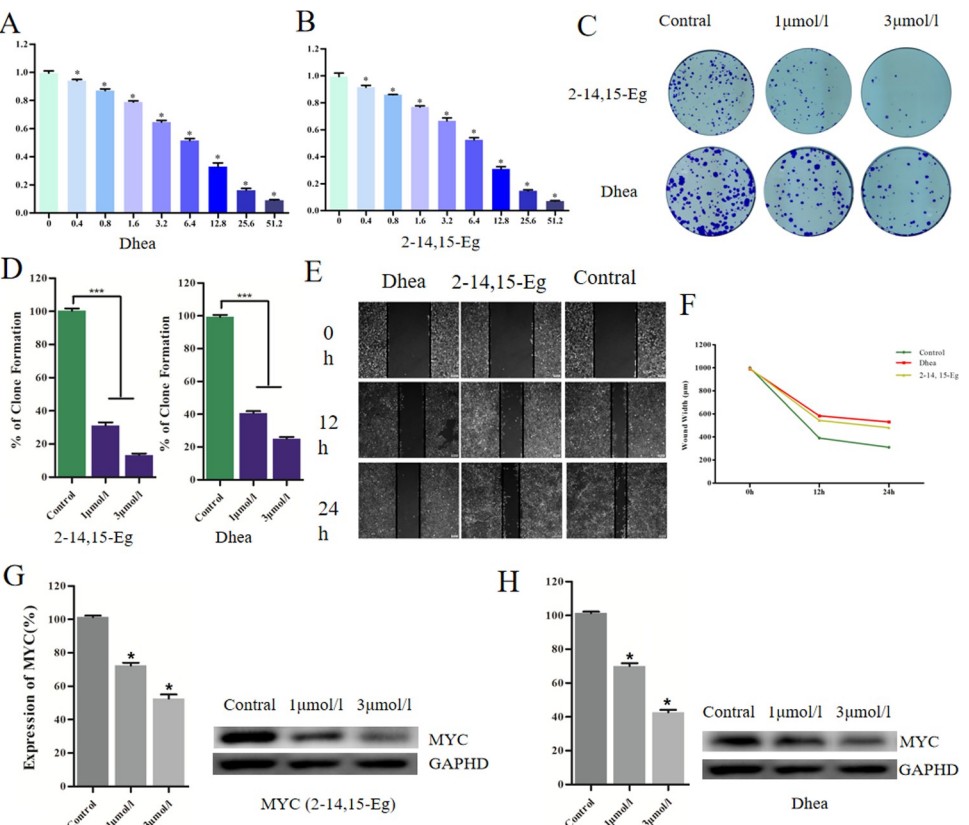

**Fig 5.** The anti-CCA effects of Dhea and 2–14,15-Eg were due to its inhibition of MYC (A) Cellular viability of cholangiocarcinoma cells treated with Dhea and (B) 2–14,15-Eg. (C) (D) Colony formation assay results of Dhea and 2–14,15-Eg anti-proliferative effects in HuCCT1 cells. (E) (F) Dhea and 2–14,15-Eg suppressing the migration of osteosarcoma cells in the scratch assay. (G) (H) Results of western blot of MYC.

associated with poorer prognosis [32]. ALB, a biomarker of nutritional and inflammatory conditions, is often used to evaluate the nutritional status of cancer patients. ALB is also associated with the prognosis of many cancers, such as oral cancer, head and neck cancer, and ovarian cancer [33–35]. Thus, higher serum ALB levels have a significant effect on the prognosis of patients with CCA after hepatectomy [36]. APOB, a key component of fat metabolism, has previously been reported that apolipoprotein levels are associated with overall cancer risk as well as breast, lung and colorectal cancer risk in men. These findings may contribute to future cancer prevention strategies [37]. IGF1, very similar to insulin in structure, is a peptide growth factor that promotes cell proliferation and inhibits apoptosis [38, 39]. Significantly increased levels of IGF1 in the biliary was reported to be related to malignant biliary obstructions, which implied IGF1 may have great effects on treatment of CCA [40]. KNG1, a cysteine proteinase, has identified as a serum biomarker for the early detection of advanced colorectal adenoma and colorectal cancer [41], as well as a potential prognostizc biomarker for oral cancer [42]. Therefore, KNG1 may exerts a enormous function on cancer. MCODE is a technique for detecting densely connected regions in huge protein-protein interaction networks that could be molecular complexes. Predicting molecular complexes is crucial because it provides another level of functional annotation. Since sub-units of a molecular complex generally function towards the same biological process, prediction of an unknown protein as part of a complex also allows increased confidence in the annotation of that protein [19]. A total of 14 modules were generated. Enriched function analysis indicated that genes in module 1 were associated

with extracellular space, extracellular exosome, and extracellular region. In module 2, the genes were mainly related in drug metabolism—cytochrome P450, chemical carcinogenesis, and metabolism of xenobiotics by cytochrome P450. At last, for module 3, the genes were enriched in fatty acid degradation, fatty acid metabolism, and metabolic pathways.

In this study, the genes ALB, MYC, APOB, IGF1 and KNG1 were demonstrated to be involved in CCA, as well as these genes may be used as potential diagnosis biomarkers, treatment targets and prognosis markers for patients with CCA. Among them, we investigated MYC in depth as a potential therapeutic target. Targeted therapies for MYC have been proposed, including the inhibition of MYC transcription, partner protein dimerization, activating post-translational modifications, and turnover [43].Studies on inhibition of MYC transcription have focused on BET inhibitors [44] and complex DNA structures called G-quadruplexes. The dimer interference mainly included MAX Inhibitor MYCMI-6, Ki-MS2-008 and so on. The post-translation regulation of MYC mainly includes the PIN1 proline isomerase, kinases that phosphorylate S62-MYC and enzymes that affect MYC ubiquitin-dependent proteolysis. However, these studies still have drawbacks. For example, although 15 different BET inhibitors are under clinical evaluation, clinical responses are limited, often leading to relapse, and inconsistent with their effect on MYC expression [45, 46].

In this study, we used LibDock, ADME/TOPKAT, CDOCKER and Molecular Dynamics Simulation, five sections of Discovery Studio for virtual screening and analysis. The compounds with higher libdock score had better energy optimization and stable conformation. According to the libdock score, the top 20 compounds were selected for further study. The pharmacological properties of these compounds were evaluated by ADME and toxicity prediction. The results showed that ZINC000008689961 and ZINC000027646625 are soluble in water and had good absorption level. In addition, they had no hepatotoxicity and were non-inhibitors of cytochrome P4502D6 (CYP2D6). Furthermore, compared with other compounds, these two compounds had lower mutagenicity, rodent carcinogenicity and developmental toxicity. To sum up, ZINC000008689961 and ZINC000027646625 were considered as safe drug candidates and further analysis were performed. Next, we studied the bonding mechanism and chemical bond of the selected candidate compounds. CDOCKER module computation indicated that the binding affinity of ZINC0000868961 and ZINC000027646625 with MYC was high. Eventually, their stabilities in natural environment was studied by molecular dynamics simulation. The results showed that the potential energy of these complexes gradually stabilizes with the passage of time. It is suggested that ZINC000008689961 and ZINC000027646625 may interact with MYC and their complexes were stable in natural environment. In opinion of all the results above, these two compounds could be developed and refined as drugs.

Afterwards, the anti-cholangiocarcinoma effects of Dhea and 2–14,15-Eg in vitro were detected by MTT assay, colony-forming assay, scratch assay and Western blotting. In MTT assay, the cellular viability in cell lines HuCCT1 decreased following the increasing drug doses of Dhea and 2–14,15-Eg. In colony-forming assay, the number and size of colony formation in Dhea and 2–14,15-Eg group were significantly lower than those in the control group, which was consistent with the results of Dhea and 2–14,15-Eg inhibiting the proliferation of CCA cells. In scratch assay, the wound width in control group decreasing as time goes by, and were smaller than in Dhea and 2–14,15-Eg group after 24h sharply which implied that Dhea and 2–14,15-Eg strongly inhibited migration of CCA cells. Our Western blotting demonstrated that MYC expression decreased with increasing drug concentrations, implying that the anti-CCA effects of Dhea and 2–14,15-Eg were due to its inhibition of MYC.

This study not only provides new ideas for the study of MYC inhibitors, but also provides mind for the development of CCA drugs. But as we all know, without thousands of

improvements and refinements, no single drug can be directly marketed. Consequently, the refinement and improvement of them is of great momentousness in the follow-up research.

## Conclusion

A total of 492 DEGs were identified in this study. GO and KEGG analysis showed that the enriched function and pathway were mainly related to extracellular exosome, P53 signaling pathway and adipocytokine signaling pathway. ALB, MYC, APOB, IGF1 and KNG1 were screened as hub genes and MYC was recognized as key therapeutic targets for CCA. ZINC000008689961 (Dhea) and ZINC000027646625 (2–14,15-Eg) were found as potent inhibitors for MYC through virtual screening technique. In series of studies demonstrated that ZINC000008689961 and ZINC000027646625 are promising and safe drug in dealing with CCA.

## Supporting information

**S1 Fig.** (A) Venn plot of mutual up-regulated and down-regulated DEGs among the three datasets. (B) Function and pathway enrichment of DEGs.
(TIF)

**S2 Fig. Schematic drawing of interactions between ligands and MYC.** The surface of binding area was added. Blue represents positive charge; red represents negative charge; and ligands are shown in sticks, with the structure around the ligand-receptor junction shown in thinner sticks. (A) ZINC000008689961-MYC complex. (B) ZINC000027646625-MYC complex.
(TIF)

**S1 Table. Functional and pathway enrichment analysis of up-regulated and down-regulated genes.**
(DOC)

**S2 Table. Functional and pathway enrichment analysis of the modules' genes.**
(DOC)

**S3 Table. MYC targeted drugs downloaded from ZICN15 database.**
(DOCX)

**S4 Table. Hydrogen bond interaction parameters for each compound with MYC.**
(DOC)

## Author Contributions

**Data curation:** Jun Kuai, Fang Yang.

**Methodology:** Lu Yang.

**Visualization:** Peisheng Sun.

**Writing – original draft:** Lei Qin, Guangpeng Li.

**Writing – review & editing:** Lanfang Zhang.

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
