## [Decision Letter · Decision Letter 0]

14 Oct 2021

PONE-D-21-25842Selected by bioinformatics and molecular docking analysis, Dhea and 2-14,15-Eg are effective against cholangiocarcinomaPLOS ONE

Dear Dr. Li,

Thank you for submitting your manuscript to PLOS ONE. After careful consideration, we feel that it has merit but does not fully meet PLOS ONE’s publication criteria as it currently stands. Therefore, we invite you to submit a revised version of the manuscript that addresses the points raised during the review process.

We look forward to receiving your revised manuscript.

Kind regards,

Nafees Ahemad

Academic Editor

Additional Editor Comments (if provided):

Dear Authors

The reviewers have suggested many improvements in the manuscript.

Journal Requirements:

3. PLOS requires an ORCID iD for the corresponding author in Editorial Manager on papers submitted after December 6th, 2016. Please ensure that you have an ORCID iD and that it is validated in Editorial Manager. To do this, go to ‘Update my Information’ (in the upper left-hand corner of the main menu), and click on the Fetch/Validate link next to the ORCID field. This will take you to the ORCID site and allow you to create a new iD or authenticate a pre-existing iD in Editorial Manager. Please see the following video for instructions on linking an ORCID iD to your Editorial Manager account: https://www.youtube.com/watch?v=_xcclfuvtxQ.

Reviewers' comments:

Reviewer's Responses to Questions

**Comments to the Author**

1. Is the manuscript technically sound, and do the data support the conclusions?

Reviewer #1: No

Reviewer #2: Yes

2. Has the statistical analysis been performed appropriately and rigorously? 

Reviewer #1: No

Reviewer #2: Yes

3. Have the authors made all data underlying the findings in their manuscript fully available?

Reviewer #1: Yes

Reviewer #2: Yes

4. Is the manuscript presented in an intelligible fashion and written in standard English?

Reviewer #1: No

Reviewer #2: Yes

5. Review Comments to the Author

Reviewer #1: The authors in their manuscript titled "Selected by bioinformatics and molecular docking analysis, Dhea and 2-14,15-Eg are

effective against cholangiocarcinoma" have attempted to to study the therapeutic effect of Dhea and 2-14,15-Eg on

cholangiocarcinoma.

Though the intention of the study is good, the results presented based on preliminary analysis does not merit publication in a journal of repute like PLOS ONE.

Reviewer #2: This is an interesting study with extensive analysis in identifying the drug candidates for cholangiocarcinoma. However, there are several issues that need to be clarified.

1. In the introduction part, the authors may provide several examples of previous studies that utilize similar approaches.

2. Several abbreviations were not defined, such as PPI, FGFR and others. Please double check.

3. All tools and databases such as MCODE, Metascape, ZINC15 database, KEGG database, etc. should be cited and provided in the reference list.

4. Materials and methods - GO and KEGG pathway enrichment analysis: what are the statistical test that were used by DAVID to identify significant GOs and pathways.

5. The figures were not cited in read order in the manuscript text. For example, Figure 4A was mentioned before Figure 1. Figure 1C were cited after figure 2a and figure 2b. The authors should reorganize on the in-text citations of the figures.

6. Several figures were not mentioned in the text. Please double check on this.

7. I think Figure 1C should be located at Figure 2, under GO and pathway enrichment analysis.

8. Please double check the Figure label of Figure 2.

9. What are the color nodes in the Figure 3 represent? Please explain this in Figure caption.

10. Each figure should have a figure title. The authors only provide the figure legends. You may refer this issue at this link https://journals.plos.org/plosone/s/figures#loc-how-to-submit-figures-and-captions.

11. 21 hub genes were mentioned in the text, but only 20 hub genes were listed in the Table 1.

12. The authors may separate the hub genes and module screening into two subsections by either adding another subsection or rename the title of this subsection (Module Screening from the PPI Network). You may also provide the total number of nodes and edges for the PPI network.

13. How many modules were generated by MCODE?

14. Among these genes, the node degree of ALB was the highest, followed by MYC with 50. This sentence is hanging. 50 of what?

15. The functional annotation and enrichment of modules genes were shown in Table. Which table?

16. Select the crystal structure of MYC as the receptor. You should rephrase this sentence.

17. It is better to provide the compounds name in the table 2, table 3 and supplementary table 3.

18. Majority of the figures and the figure labels are not clear.

19. In the discussion part, the authors may discuss the importance of hub genes in human diseases.

20. What is the purpose of performing module selection? Is MYC detected in the modules? Can you discuss this result in the discussion?

6. PLOS authors have the option to publish the peer review history of their article (what does this mean?). If published, this will include your full peer review and any attached files.

Reviewer #1: No

Reviewer #2: No

---

## [Author Response · Author response to Decision Letter 0]

30 Oct 2021

Dear Editors:

Thank you for the valuable comments of our manuscript entitled “Selected by bioinformatics and molecular docking analysis, Dhea and 2-14,15-Eg are effective against cholangiocarcinoma”. We have studied the comments carefully and have made corrections in the manuscript which we hope to meet with approval. The main corrections in the paper and the responses to editor’s comments are as following.

Review Comments to the Author

Reviewer #1: The authors in their manuscript titled "Selected by bioinformatics and molecular docking analysis, Dhea and 2-14,15-Eg are effective against cholangiocarcinoma" have attempted to to study the therapeutic effect of Dhea and 2-14,15-Eg on cholangiocarcinoma.

Though the intention of the study is good, the results presented based on preliminary analysis does not merit publication in a journal of repute like PLOS ONE.

Response：

Thanks for your comments, but I don't agree with you someway. Natural molecules have contributed significantly to not only molecular biological research but also potential drug development. A number of relevant studies have been published in reputable journals. Besides, we carried out this research with great care and precision and MTT assay, colony-forming assay, the scratch assay and Western blotting were performed to verify our findings. I hope you could reconsider it again, and your opinion is very important to us. Thank you very much.

[1] Li, W., et al., Ten-gene signature reveals the significance of clinical prognosis and immuno-correlation of osteosarcoma and study on novel skeleton inhibitors regarding MMP9. Cancer Cell Int, 2021. 21(1): p. 377.

[2] Zhong, S., et al., Selected by gene co-expression network and molecular docking analyses, ENMD-2076 is highly effective in glioblastoma-bearing rats. Aging (Albany NY), 2019. 11(21): p. 9738-9766.

Reviewer #2: This is an interesting study with extensive analysis in identifying the drug candidates for cholangiocarcinoma. However, there are several issues that need to be clarified.

1. In the introduction part, the authors may provide several examples of previous studies that utilize similar approaches.

2. Several abbreviations were not defined, such as PPI, FGFR and others. Please double check.

3. All tools and databases such as MCODE, Metascape, ZINC15 database, KEGG database, etc. should be cited and provided in the reference list.

4. Materials and methods - GO and KEGG pathway enrichment analysis: what are the statistical test that were used by DAVID to identify significant GOs and pathways.

5. The figures were not cited in read order in the manuscript text. For example, Figure 4A was mentioned before Figure 1. Figure 1C were cited after figure 2a and figure 2b. The authors should reorganize on the in-text citations of the figures.

6. Several figures were not mentioned in the text. Please double check on this.

7. I think Figure 1C should be located at Figure 2, under GO and pathway enrichment analysis.

8. Please double check the Figure label of Figure 2.

9. What are the color nodes in the Figure 3 represent? Please explain this in Figure caption.

10. Each figure should have a figure title. The authors only provide the figure legends. You may refer this issue at this link https://journals.plos.org/plosone/s/figures#loc-how-to-submit-figures-and-captions.

11. 21 hub genes were mentioned in the text, but only 20 hub genes were listed in the Table 1.

12. The authors may separate the hub genes and module screening into two subsections by either adding another subsection or rename the title of this subsection (Module Screening from the PPI Network). You may also provide the total number of nodes and edges for the PPI network.

13. How many modules were generated by MCODE?

14. Among these genes, the node degree of ALB was the highest, followed by MYC with 50. This sentence is hanging. 50 of what?

15. The functional annotation and enrichment of modules genes were shown in Table. Which table?

16. Select the crystal structure of MYC as the receptor. You should rephrase this sentence.

17. It is better to provide the compounds name in the table 2, table 3 and supplementary table 3.

18. Majority of the figures and the figure labels are not clear.

19. In the discussion part, the authors may discuss the importance of hub genes in human diseases.

20. What is the purpose of performing module selection? Is MYC detected in the modules? Can you discuss this result in the discussion?

Response：Those comments are all valuable and very helpful for revising and improving our paper, as well as the important guiding significance to our researches. We have studied comments carefully and have made correction which we hope meet with approval.

1.The similar approaches had been provided in the introduction part.

2.Thanks for your comment. The all abbreviations were provided.

3.All tools and databases had been cited and provided in the reference list.

4. In DAVID, Fisher's Exact test is adopted to measure the gene-enrichment in annotation term, and P < 0.05 and counts > 2 were set as the threshold values. Significant GOs and pathways is filtered by this threshold.

5.The order of figures had been revised.

6.Thanks for your comment. We had checked it and mentioned all figures in this manuscript.

7.We had revised it.

8.The Figure label of Figure 2 had been revised.

9.The color nodes of Figure 3 had been explained in figure captions. The degrees of nodes were labeled in different colors, the red color means bigger degrees.

10.The figure titles had been provided.

11.We had checked it and corrected the number.

12.We had revised this part as your comment, and the total number of nodes and edges for PPI network also been provided.

13.There are 14 modules were generated by MCODE.

14. We had revised this sentence.

15. The table was provided.

16. We had revised this sentence.

17.The compounds name in the table 2, table 3 and supplementary table 3.

18.The figures and the figure labels are revised, and 300dpi figures were provided.

19.The importance of hub genes in human diseases were discussed in discussion section.

20. MCODE is a technique for detecting densely connected regions in huge protein-protein interaction networks that could be molecular complexes. Predicting molecular complexes is crucial because it provides another level of functional annotation. Since sub-units of a molecular complex generally function towards the same biological process, prediction of an unknown protein as part of a complex also allows increased confidence in the annotation of that protein. Performing module selection and the result were discussed. MYC is also detected in the modules.

Dear Editors,

Thank you for your work dealing with this manuscript, thank you very much. If there is any problem, please don’t hesitate to contact us. I will reply you as soon as possible. Best wishes to you. May its blessings lead into a wonderful year for you and all whom you love.

---

## [Editor Report · Decision Letter 1]

4 Nov 2021

Selected by bioinformatics and molecular docking analysis, Dhea and 2-14,15-Eg are effective against cholangiocarcinoma

PONE-D-21-25842R1

Dear Dr. Li,

We’re pleased to inform you that your manuscript has been judged scientifically suitable for publication and will be formally accepted for publication once it meets all outstanding technical requirements.

Kind regards,

Nafees Ahemad

Academic Editor

PLOS ONE
---

## [Editor Report · Acceptance letter]

21 Jan 2022

PONE-D-21-25842R1 

Selected by bioinformatics and molecular docking analysis, Dhea and 2-14,15-Eg are effective against cholangiocarcinoma 

Dear Dr. Li:

I'm pleased to inform you that your manuscript has been deemed suitable for publication in PLOS ONE. Congratulations! Your manuscript is now with our production department. 

Kind regards, 

on behalf of

Dr. Nafees Ahemad 

Academic Editor

PLOS ONE